# Facilitators, barriers, and strategies for the implementation of peer-led tuberculosis active case finding among people who use drugs in Dar es Salaam, Tanzania

Lilian Tina Minja[1,2,3]*, Liana Monica Minja[1], Kilian Mlalama[1], Doreen Pamba[1], Jerry Hella[4], Samwel Likindikoki[5], Cassian Nyandindi[6], Jessie Mbwambo[7], Klaus Reither[2,3], Jennifer M. Belus[3,8]

1 National Institute for Medical Research, Mbeya, Tanzania, 2 Swiss Tropical and Public Health Institute, Allschwil, Switzerland, 3 University of Basel, Basel, Switzerland, 4 Ifakara Health Institute, Dar es Salaam, Tanzania, 5 Muhimbili University of Health and Allied Sciences, Tanzania, 6 Drug Control and Enforcement Authority 7 Muhimbili National Hospital, Dar es Salaam, Tanzania, 8 University Hospital Basel, Basel, Switzerland.

* liliantinaminja@gmail.com

## Abstract

### Background

Globally, tuberculosis (TB) is the leading cause of death from a single infectious agent. In 2023, an estimated 2.7 million cases of TB were undiagnosed or unreported. To address missing cases, the World Health Organization recommends systematic screening for TB. This is synonymous to active case finding (ACF) and involves provider-initiated screening and testing for TB. Despite the high incidence and prevalence of TB among people who use drugs (PWUD), there is a significant gap in data, on their perspectives, regarding the implementation of TB ACF services. This study aimed to explore facilitators and barriers to implementing peer-led TB ACF, as perceived by both, current and potential service users.

### Methods

We conducted in-depth interviews among purposively selected adult PWUD in Dar-es-Salaam region, Tanzania. Study participants included: (1) peer PWUD with prior history of illicit drug use and medication-assisted treatment (MAT) (n=10), (2) current medication-assisted treatment service users receiving clinic-based daily methadone (n=8), and (3) community PWUD not on MAT recruited from various community locations (n=4). All peer PWUD were experienced in TB ACF. Thematic content analysis was utilized with the support of NVivo12.

### Results

Our findings are presented into two categories: individual and structural, with three main themes pertaining to peer-led TB ACF: (1) facilitators (2) facilitators for targeted

**Data availability statement:** Some interview transcripts include identifiable and sensitive information, and publicly releasing the data could compromise participant confidentiality. To protect participant privacy, the data generated and analyzed during this study is not publicly accessible. The data generated and analyzed can however be made available upon reasonable request, provided that the data sharing agreement is approved by the Medical Research coordinating Committee in Tanzania. For data availability requests, please contact info@nimr.or.tz

**Funding:** This work was supported by the Swiss Tropical and Public Health Institute, Tanzania National Institute for Medical Research – Mbeya Centre and the Ifakara Health Institute. Received award - LTM The funders had no role in study design, data collection and analysis, decision to publish, or preparation of the manuscript.

**Competing interests:** The authors have declared that no competing interests exist. The funders had no role in the study design, data collection and analysis, decision to publish, or preparation of the manuscript. JMB is supported by the Swiss National Science Foundation Ambizione grant [PZ00P1_201690; PI: Belus]

improvement and optimization and (3) barriers. A critical facilitator was the acceptability of peer PWUD in providing TB ACF services. Key facilitators for targeted improvement and optimization included the TB screening tool, mobile TB diagnostic services, integrated methadone/TB services, and monetary incentives to peer PWUD. Barriers included inadequate adherence to infection prevention and control (IPC) measures when providing TB ACF services resulting in a reluctance to wear face masks due to stigma, misconceptions that prior TB preventive therapy among peers negates their need for continued IPC adherence, high mobility of PWUD and the fear of withdrawal symptoms associated with the use of anti-TB medication. Due to this fear, many PWUD preferred not to take anti-TB, as they were concerned about the potential severity of withdrawal symptoms.

## Conclusion:

Our findings highlight the crucial role of peer-led approaches in enhancing TB ACF among PWUD. Peer acceptance as service providers highlights the potential of community-driven interventions. Strengthening facilitators and addressing challenges is key to optimizing these services. Future research should explore the feasibility of providing peer-supported TB diagnosis and treatment services at friendly drop-in centers.

## Recommendations:

1. Strengthen mobile diagnostic services by increasing their frequency and coverage, enabling timely diagnosis and treatment.
2. Enhance the TB symptom screening tool by including a symptom-independent test, such as a chest X-ray, as the symptoms of illicit drug use can mask and mimic TB symptoms making diagnosis challenging.
3. Address stigma and misconceptions through peer-led education and awareness campaigns that utilize audio-visual materials tailored to PWUD. This will promote adherence to IPC measures and create a more supportive environment for TB ACF activities.
4. Use anti-TB with minimal interactions with opiates or shorter TB treatment regimens to prevent withdrawal symptoms and improve adherence.

---

### Introduction

Globally, tuberculosis (TB) is again the leading cause of death from a single infectious agent. In 2023, there were an estimated 2.7 million undiagnosed TB cases worldwide [1]. Tanzania is among the 30 high TB burden countries globally, with an estimated TB incidence of 183 cases per 100,000 population (1). Despite a significant 14% increase in the TB case detection rate from the previous year in 2022, the TB case notification gap widened in 2023, with approximately 1,700 more cases left unreported. As a result, 28,750 TB cases remained undiagnosed or unreported in 2023 [2–4].

An estimated 292 million people worldwide were reported to use drugs, including 60 million in Africa in 2021 [5]. The number of PWUD in Africa is projected to rise by 40%, significantly outpacing the projected global increase of 11% by 2030. East Africa is anticipated to experience the sharpest growth [6–8]. Tanzania, located in East Africa is a transit country for illicit drugs and had an estimated 300,000 PWUD according to the latest report in 2014 [9]. With the increased interception of opiates in the country in 2020, the number of PWUD is likely higher [10].

In the fight to end TB, the World Health Organization (WHO) has endorsed systematic screening for TB as a key component of the first pillar in its strategy to end the disease [11]. Systematic screening also known as active case finding is predominantly provider driven, and involves the active identification of people at risk for TB disease in specific target groups, through the use of screening tests, examinations, or other procedures that can be rapidly applied [11]. A positive screening result will require diagnostic tests and clinical assessments for disease confirmation. When ACF is implemented in health facilities, it is termed intensified ACF, while in the community, it is known as enhanced ACF [11,12]. In enhanced ACF a diagnostic sputum sample can be collected at the screening site, such as a dwelling or gathering place, or at a mobile diagnostic van. Alternatively, the presumptive TB case may be referred to a health facility for confirmatory testing. There are numerous documented benefits to ACF, in comparison to passive case finding, including early TB case detection and treatment initiation, improved TB treatment outcomes, reduced patient costs per TB diagnosis, initiation of TB preventive therapy among eligible individuals, and ultimately reduced TB transmission and reduction in TB prevalence [13–15].

Given these benefits, the WHO strongly recommends that ACF be conducted among high-risk groups, including people living with HIV, contacts of people known to have TB, people exposed to silica, and in prisons and penitentiary institutions [11]. PWUD have a heightened vulnerability to TB due to multiple overlapping risk factors which include co-infection with HIV and hepatitis, a greater likelihood of incarceration, exposure to TB patients, smoking, alcohol use disorders and malnutrition [16–24]. An integrated bio-behavioral survey (IBBS) conducted among people who inject drugs (PWID) in Dar es Salaam, Tanzania in 2017 reports an HIV prevalence of 8.7% and an HIV/HCV co-infection prevalence at 43.1% [17,21]. The HIV prevalence of 8.7% among PWID is a notable decrease from 15.5% in 2015 [18], however, this figure is nearly double the national HIV prevalence of 4.4% [25]. The IBBS, also found that only 6.5% of the 611 participants reported enrollment in MAT clinics [21] highlighting the urgent need for community-based interventions in TB ACF among PWUD.

Studies on TB ACF in low- and middle-income countries, primarily conducted in the general population, have identified key barriers such as limited access to testing supplies, stigma and discrimination associated with TB diagnosis, long travel distances to diagnostic facilities causing delays, low community awareness, inadequate healthcare training, staff shortages, and language barriers. Reported facilitators include, reaching vulnerable populations, incentivization and conducive service environment [26–29]. Despite the high incidence of TB among PWUD, there is a significant gap in data on their perspectives regarding TB ACF. This study aimed to explore facilitators and barriers to implementing peer-led TB ACF, as perceived by both current and potential service users.

## Materials and methods

### Study setting and design

This was an exploratory qualitative study conducted among PWUD in Dar es Salaam region in Eastern Tanzania. Dar es Salaam was selected as it is the most populous region in Tanzania, with an estimated 5.4 million inhabitants [30] and accounts for the highest TB notified cases in the country, with 17% in 2023 [4]. Participants were recruited from three different MAT clinics located in Ilala, Kinondoni and Temeke districts and through community and non-governmental organizations supporting TB activities among PWUD in the same region. The selected MAT clinics were the first clinics to provide methadone to PWUD clients in the country in 2011 [31]. Currently, there are eleven clinics located in different regions of the country and serve about 10,000 clients. The clinics provide daily methadone free of charge as a directly observed

therapy, with some clients receiving take-home doses. TB ACF is currently implemented through trained peers at the MAT clinics and in the community.

## Recruitment and sample selection

This study enrolled adult PWUD who were 18 years and above, willing to partake in a qualitative face to face interview, have their interview audio recorded, and able to communicate in Swahili or English. Purposive sampling was used to recruit three different groups of participants (1) peer PWUD with a prior history of illicit drug use and medication-assisted treatment completion (n = 10), (2) current medication-assisted treatment service users receiving clinic-based methadone (n = 8), and (3) community PWUD recruited from various community locations (n = 4). All peer PWUD interviewed were experienced in providing TB ACF and were selected from the MAT clinics, based on their good conduct and positive progress at the MAT clinics. Some peer PWUD would additionally be involved in other harm reduction services such needle syringe program and HIV services. The exclusion criterion was untreated mental illness that could interfere with study participation, which was based on review by MAT clinic managers who are psychiatrists by profession.

Sampling was conducted through lists provided by the MAT clinic managers taking into consideration; HIV and TB status, being a current or prior MAT client and involvement in ACF activities (peer PWUD). Community-based organizations were selected based on their focus area (TB, HIV, and/or substance use) and provision of harm reduction services that linked clients to MAT clinics. Through community-based organizations, community PWUD were linked to the study team and invited to participate.

## Data collection

Semi-structured interview guides were developed to assess the facilitators and barriers of ACF for TB among PWUD. Data collection for the study took place from 23/01/2023–01/03/2023. The interview guides were piloted on two peer PWUD and one MAT client. Feedback from pilot interviews indicated that the questions were well understood and no substantive changes were needed in the interview guides. However, during subsequent interviews, questions in the interview guides were rephrased or added depending on the participants responses and probing skills. The interview guide for peer PWUD differed from the interview guide for MAT and community PWUD in that the peer PWUD guide included questions on the TB ACF implementation process (e.g., screening tools, diagnosis, and referral process). The final interview guides used for the study are attached [S1 File. Interview guides - English and Swahili]. Demographic information on age, sex, level of education, occupation, HIV status among prior or current MAT PWUD, previous history of TB were collected. A total of 24 participants were deemed eligible and were provided with information about the study and invited to participate. Two participants did not provide consent - one reported not to have time, the second participant did not want to be audio recorded. For consistency in the level of detail that could be extracted from participants' interviews, we wanted to ensure that all interviews could be captured verbatim, requiring audio recording. For this reason, it was included as a study inclusion criterion, which was approved by the ethics board and stipulated in the informed consent form. Thus, the participant was not enrolled. In-depth interviews were conducted with 22 participants who provided written informed consent. Three experienced team members (LTM, DP, KM), two of which are women conducted the interviews in Swahili, the local language. LTM is a medical doctor with a master's degree in medicine. DP has a bachelor of arts in sociology and master's degree in international health. KM has a bachelor of arts in sociology and a master's degree in public health research. All interviewers have 5–10 years' experience in medical research and work as research scientists. After being invited to participate in the study and providing informed consent, participants were asked if they were comfortable and willing to be interviewed by the interviewer, regardless of gender. None of the interviewed participants expressed any gender preferences. Interviews were conducted in a private room at the local MAT clinic or at the partner community-based organization to ensure confidentiality. All interviews were audio recorded and field notes taken. In-depth interviews lasted an average of

40 minutes (range 20–60 min). Data collection followed guidelines for qualitative research sample size, focusing on data saturation and information power. We reached these targets in our data collection, with the team observing repetition of information towards the end of the interviews. This ensured meaningful insights to be drawn from the study.

### Data analysis

Thematic analysis was used to analyze the qualitative data using NVivo 12 software. The audio recordings were transcribed verbatim in Swahili before translation into English. The data was translated by a professional translator fluent in both the local language Swahili and English. The translator was familiar with the context of the study to ensure accuracy and cultural relevance. The translated transcripts and field notes were further double checked for accuracy by the research team (LTM, DP, KM) thus also allowing for familiarization of data. In order to develop initial draft codebooks to guide subsequent coding, LTM, DP and KM randomly selected one transcript from each of the three participant groups thus, each with a transcript to work on. Deductive codes were applied based on research questions and few others were emergent from the transcripts. Thereafter, the team met for a group review of the coded transcripts. During the review, the team extensively reviewed the codebooks, discussions were engaged and came to an agreement through discussions when discrepancies were noted. Thus, certain codes were deleted, merged and rephrased. The final codebooks were subsequently applied to the rest of the transcripts which were randomly divided among the team. Thereafter, codes were grouped into categories and themes were developed from categories with similar meaning.

**Ethical considerations.** The study received approvals from the National Health Research Ethics Review Committee and Ifakara Health Institute Institutional Review Board in Tanzania. The investigator welcomed the participant or legally acceptable representative (for illiterate participants) to participate. The investigator then provided the participant with an information and informed consent form, providing information on the study, including its potential benefits and risks. Adequate time was provided to the participant to read and ask questions about the study, and an assessment of understanding was conducted by the investigator to confirm the participant had understood the information provided. The language used was as non-technical as possible. Written informed consent was obtained from all study participants. In the case of illiteracy, informed oral consent was attested by an impartial witness and documented with the patient's fingerprint. This study followed the COnsolidated criteria for REporting Qualitative research [32] checklist **[S2 File. COREQ checklist]**.

## Results

Study participants had a mean age of 42.5 years (SD = 6.96; range 28–57 years). The majority of participants were male (64%) and 60% had completed primary level education. All peers (n = 10) received financial support as a monthly incentive for their role as peer PWUD and considered themselves as employed. Among the remaining 12 participants (MAT clients and community PWUD), 58% were unemployed while, 42% had informal occupations. Previous history of TB was reported in 41% of all interviewees. The HIV status was known for 18 participants, with eleven being HIV negative and seven HIV positive. Our findings are presented under three themes: (1) facilitators (2) facilitators for targeted improvement and optimization and (3) barriers to peer-led TB ACF among PWUD. These themes are broadly categorized into structural and individual factors [S3 Fig].

### Facilitators

**Peer provision of TB ACF services.** Peer PWUD play a key role in ACF by being the link between the health facility and PWUD residing in the community. Provision of ACF by peer PWUDs is done at the community level or at the MAT clinic. Peers felt accepted by community PWUD because they had experienced similar situations of illicit drug use in the past, making them familiar with the environment and behaviors. This sentiment was echoed by community PWUD, who

felt seen, and that health services were being taken to them by non-discriminating fellow peers. This led to an open and friendly interaction with PWUD agreeing to be screened.

*"… I trust peers… because they are also like me… after being educated they have changed, and when they tell me about changing, I must believe him because they are talking through experience."* (Male, 47 years, community PWUD)

### Facilitators for targeted improvement and optimization

**TB screening tool.** The availability of a TB symptom screening questionnaire, which asks about the presence or absence of common TB symptoms, such as cough, fever, night sweats and weight loss was perceived as easy to use and helpful in making their job manageable. Once a client reports a TB symptom, a sputum sample would be collected and taken to be tested and results returned by the peer. Peers would track those with positive results and escort them to a facility for registration and treatment initiation. When faced with negative sputum results in symptomatic clients, the peers would advise the client to do a chest x-ray and visit a health facility for further management.

*"We start talking with our clients about TB symptoms that are listed on the TB form and ask them if they have any TB symptoms that we have mentioned, if they report TB symptoms, we tick the symptom and give them a sputum container to provide a sputum sample, the form is easy to use." (*Male, 41 years, peer PWUD).

Since TB symptoms overlap with those of drug withdrawal and the use of illicit drugs can mask TB symptoms, PWUD would have doubts on whether the symptoms that they have are TB related or not. Peers raised concerns about the reliability of the screening tool, as they would report finding clients with several TB symptoms but testing negative on sputum samples

*"This is a big problem as when PWUD start experiencing TB symptoms, at times they think maybe it is withdrawal symptoms while it is TB affecting them…."* (Female, 39 years, peer PWUD).

*"They tested my sputum sample and told me I do not have TB, I continued being sick, later on I went back and was told that I have TB and was started on treatment."* (Male, 42 years, peer PWUD).

### Mobile TB diagnostic services

The availability of mobile TB diagnostic services, which were accessible to PWUD during ACF services, was highlighted as an important facilitator. The mobile van would be stationed in open areas such as a sports pitch. Peer PWUD would identify individuals with presumptive TB in the PWUD galleries and accompany them to the mobile van for spot sputum sample collection. Sputum samples were tested and results provided on the same day. Free chest x-ray services were also provided when needed. Those diagnosed with active TB would receive an initial dose of TB treatment be escorted by the peer to a health facility or MAT clinic, where provision of TB treatment and other services would continue. This, ensured timely TB diagnosis and treatment initiation.

*"… to be honest TB is a very big problem, our only hope is in the mobile van which helps in collecting sputum samples that are tested immediately, and you can also do a chest x-ray, this has been a big help to us and many people visit the mobile diagnostic van and get diagnosed quickly."* (Female, 39 years, peer PWUD)

However, shortcomings in the mobile van services were identified, including infrequent and inconsistent operations. The services were highly sporadic, primarily conducted on special occasions such as World TB and World AIDS days making it difficult to predict where they would be held. Currently, their frequency was reported to drop to just 2–3 times per

year. In the absence of the mobile diagnostic van, majority of participants reported difficulties in seeking health services. This included transport costs to attend a health facility that provided TB services. Furthermore, participants reported long waiting times for testing and delays in obtaining sputum results.

*"They fear the costs involved [TB diagnosis and treatment], so they think it is better to stay in the galleries while TB continues to affect them"* (Male, 39 years, peer PWUD)

*"… I see others providing sputum samples [at a health facility] but when I see them, I leave, I don't have time because it takes too much time… for some they get their results within 3 days, others 2–3 weeks, and others it takes up to a month until they get their results…"* (Male, 28 years, community PWUD).

If a symptomatic presumptive TB client tests negative on the sputum sample, they would be advised and required to have a chest x-ray performed, with most unable to afford the cost.

*"I did not have money to pay for an x-ray (chest x-ray), you know I provided a sputum sample and was told to do an x-ray, but I did not have the money to pay for the x-ray… I did not have any money"* (Female, 37 years, MAT client)

### Integrated health services

In addition to provision of daily methadone, MAT clinics also conduct TB ACF, test and counsel for HIV, and provide anti TB and or anti-retroviral therapy for HIV and AIDS as needed. Having these services integrated under one roof was convenient and importantly, peer PWUDs were identified through the MAT clinics.

**"***… this new client has been started on methadone today, but they must also be investigated for TB... we have our processes here, when you start methadone treatment you must be investigated for TB disease."* (Female, 32 years, MAT client).

Despite being perceived a facilitator, interviewees felt that the integrated health services could be improved and made more user friendly. For instance, one participant mentioned that the designated area within the MAT clinics where HIV testing and treatment services were being provided lacked adequate privacy. Another participant reported that they took their ART at a different facility from their daily methadone treatment to avoid facing stigma.

*"This place is not okay… if it is possible to have a room somewhere else where they cannot see you taking other drugs [ART]... Many times, they [health care workers] make us sit outside and then they start calling us, one after another."* (Male, 43 years, MAT client).

*"But I don't take my medication [ART] here, I attend another facility where I take my medication [ART]. You know people who use drugs have a tendency of stigmatizing each other, people start talking about you."* (Female, 39 years, peer PWUD)

### Monetary incentive to peer PWUD

Provision of monthly monetary incentives to peers, funded by non-governmental organizations was recognized and reported as a facilitator. This compensation covered peer PWUD's time and effort while also motivating them to continue with TB screening activities.

*"It enables us to go to the galleries when seeking community PWUD… at the end of the month we get a bit of money as an incentive"* (Female, 42 years, peer PWUD)

However, due to high mobility of PWUD, peers had to visit multiple galleries spread over long distances to conduct TB screening activities. And, although the monthly incentive was not intended for transport, the long distances led them to use a significant portion of it to cover bus fares and travel expenses. At times they had no money, which hindered their ability to carry out ACF activities. Due to this they reported that the monthly incentive was not enough.

*"... that is why they need to improve our incentive as this work is not easy and it is risky, we need to travel and move a lot, but at the end of the month we receive…, this is not enough."* (Male 42 years, peer PWUD).

### Barriers

**Highly mobile population.** Identifying presumptive TB cases among community PWUD and ensuring that they are properly investigated and linked to health care is a difficult task. In the absence of consistent mobile diagnostic facilities, spot sputum samples collected in the community are taken by the peers to selected health facilities for testing. Peers would then trace the results and predominantly communicate the positive TB results to the community PWUD with confirmed TB disease, or negative results among the ones they perceived were sicker and link them to health care. However, tracing community PWUD diagnosed with TB was challenging, often taking several days to locate them or, in some cases, failing to find them altogether. This was because community PWUD frequently move within and between communities and lack permanent settlements. This disrupted the continuity of timely healthcare access.

*"… PWUD are highly mobile, moving from one dwelling to another, you may find them in one place today, the next day when you get the sputum results [positive] you find they have moved, they don't stay in one dwelling…".* (Female, 57 years, peer PWUD)

### Non adherence to infection prevention and control measures

Wearing of face masks as one of the infection prevention and control (IPC) measures was felt by peer PWUD to be stigmatizing to the person being screened. Peer PWUD felt that mask free interactions were more personal and encouraged participation. As a result, peer PWUD would often not wear face masks. Some peer PWUD would report that they improvise on the infection prevention and control measures, such as by assessing the direction of the wind or air flow and trying to ensure the flow of air was blowing away from the peer. Others believed that because they had taken TB preventive therapy, they were protected from getting active TB disease and would thus not take any IPC measures. These adaptations to IPC measures and or misconceptions to TB preventive treatment protectiveness may have increased the risk of them acquiring TB disease multiple times

*I have had TB twice, both times while I was a community outreach worker, this is due to the environment we work in, when we visit the PWUD galleries you find many people sitting together in rooms that do not even have enough air, you cannot wear a mask as they will see you are stigmatizing them. So, you have to pretend that you are all the same, so that they invite you in, after talking to them and they agree to provide a sputum sample you find out that they have TB."* (Female, 57 years, peer PWUD)

*"I like helping people, I cannot wear a mask from morning to evening, that will be a lie. But I try to take precautions, for example I took six months of TB preventive therapy"* (Female, 39 years, peer PWUD)

Interestingly, when mandated to wear a face mask at the MAT clinic during daily methadone visits, both the clients and peers conducting TB screening would wear face masks. Notably though, sharing of face masks was reported by MAT

clients, in that as one leaves the MAT clinic the mask would be passed on to the next client entering. Knowing this was not safe they would turn the mask inside out as they re-wore it. As reusable free face-masks were sometimes provided at the clinics, they reported losing them and preferred to use their money to buy things other than face masks. At one clinic the face masks would be numbered by a unique number, this prevented face mask sharing, however may have discouraged wearing of the face masks, as the face masks looked different and the numbers identified them as PWUD.

*"…You need to buy your own face mask… we rather share a mask with four up to five people, that is why we infect each other with TB, someone takes their methadone and at the gate they give their mask to another person who turns it inside out and wears it."* (Female, 33 years, peer PWUD)

### Fear of withdrawal symptoms attributed to anti-TB

Fear of developing withdrawal symptoms was an emerging barrier attributed to the interaction of methadone and TB drugs, PWUD would notice that they start developing withdrawal symptoms when taking TB treatment. This would result in them either stopping or skipping their TB treatment, thus not adhering to anti TB. This may have also discouraged sputum testing in fear of what may happen if they are diagnosed to have TB disease.

*"They face challenges, like when taking methadone and TB drugs, the drug [illicit drug] effect goes down... TB drugs are stronger than methadone. They do not use the TB tablets they are given but throw them away and stop treatment"* (Female, 31 years, MAT client)

## Discussion

We have observed three major themes that describe (1) facilitators (2) facilitators for targeted improvement and optimization and (3) barriers to peer-led TB ACF among PWUD. The themes fall broadly under two main domains, that is individual and structural. A key facilitator was the acceptability of peer PWUD in provision of TB ACF services. Facilitators identified for targeted improvement and optimization included, the TB screening questionnaire, mobile diagnostic services, integrated TB ACF with MAT services including HIV care and treatment, and monetary incentives. Barriers included high mobility of PWUD, stigma attributed to adherence to IPC measures, and fear of developing illicit drug withdrawal symptoms when on anti-TB. This was the first study, to our knowledge, to explore perceptions of PWUD to peer led-TB ACF in Dar es Salaam, Tanzania.

A key facilitator of TB ACF was the acceptability of peer PWUD in providing these services. Our findings are similar to a study among the urban poor in Cambodia, which highlighted the benefits of expanding community networks through peer involvement and emphasized the importance of utilizing peers due to their deep understanding of their communities [33]. Another study also reported an increase in TB ACF cases facilitated by peer engagement [26], however, peer volunteers expressed the need to be accompanied by a trained health worker, as this enhanced their credibility and respect within the community [33]. While our study did not specifically report this, it raises the question of whether the presence of a trained health worker could also improve adherence to infection prevention and control (IPC) measures.

Extensive literature highlights the limitations of TB symptom screening in the general population, including the subjectivity of both the interviewer and the individual being screened. Studies highlight that approximately 50% of TB cases are asymptomatic, and among those who do experience symptoms, some may not report them, thus advocating for screening tools that go beyond symptoms [34]. Furthermore, a recent review looking at the clinical utility of the WHO-recommended four symptom screening tool and clinical prediction models among PLHIV reports suboptimal utility of the symptom screen tool [35]. Among people who use drugs (PWUD), TB symptoms may be further masked due to the physiological effects of

illicit drug use [36], reinforcing concerns about the reliability of symptom screening in this population. Given these limitations, there is a strong need to incorporate symptom independent screening tools such as a chest x-ray or C-reactive protein. Since, 2021 the WHO recommends the use of computer aided detection (CAD) software such as CAD4TB in those 15 years and older for detection of TB in chest x-rays. Several studies report an increase in case detection, with the potential of reducing costs incurred by expensive microbiologic tests [37]. Thus, important to evaluate its performance among PWUD.

Studies on AFC in the general population have shown that services provided in the community (often in people's homes) were beneficial in avoiding out of pocket costs incurred in the diagnosis process [28,38] and the presence of mobile diagnostic vans limited attrition during diagnosis and increased yields in targeted populations [13,39]. This underscores the importance of optimizing mobile diagnostic services in coverage, frequency, and consistency. To ensure the sustainability of such services, it is essential to assess the running costs of mobile diagnostic vans, including expenses for fuel, maintenance, and incentives for service providers, while exploring cost-effective and sustainable options. Additionally, integrating TB diagnosis and treatment services into easily accessible drop-in centers or community-based organizations that cater to PWUD could be considered. Drop-in centres do not currently offer TB diagnostic services, however act as a vital link between community PWUD and health facilities. This approach has the potential to reduce attrition rates in TB services by providing screening, diagnosis, and treatment in one location.

Our study findings indicate that PWUD can perceive the wearing of face masks as stigmatizing. This results in inconsistent use, compromising TB transmission prevention efforts.

The country's TB infection control measures recommend wearing of surgical or procedure masks in confirmed infectious or presumptive TB patients and respirators for health care workers [40,41]. Research specifically on PWUD and wearing face masks for infection prevention are lacking. However, wearing face masks has generally been associated with stigma and discrimination due to their association with TB disease and or the subsequent association of TB disease with HIV and AIDS [42]. In our study, wearing of face masks by peer PWUD in the community was not often accepted. This could be due to the fear of social judgement as face masks are often associated with protecting the user among health service providers [43], implying that the person being screened is infectious. On the other hand, a person with TB disease is advised to wear a face mask in public, thus implying that possibly the wearer has TB disease. Additionally, face masks provided at MAT clinics sometimes differed from those used in the general community. Thus the color, material, type of face mask or certain identifiers on it could hinder its use, as one might identify the wearer as a PWUD [44]. Nonetheless, studies from COVID-19 suggest that uptake of wearing face masks can be enhanced by strengthening the belief of solidarity and promotion of positive social messages [45,46].

Fear of developing withdrawal symptoms during TB treatment was noted as an emerging theme. Rifampicin, a drug used in TB treatment is known to decrease the effect of opioids due to the association with cytochrome P enzymes necessitating opioid dose adjustment or optimization of methadone dose [47,48]. Optimization of methadone may be feasible among TB patients attending MAT clinics, however challenging with community PWUD not on methadone. Rifabutin can be considered as an alternative to rifampicin in TB treatment, as it has less interaction with methadone and is not associated with withdrawal symptoms [47,49,50].

This study has numerous strengths, including the ability to gather insights from three different groups of PWUD, that is PWUDs in the community, those attending MAT clinics and on methadone, and also PWUD who have been successfully weaned off illicit drugs. This allowed for a broader understanding of their experiences and perspectives. In addition, partnering with the country's first three MAT clinics provided an opportunity to engage with experienced as well as new clients newly enrolled into MAT clinics and gather practical, real-world information on the topic. The study has several limitations. First, it was conducted in a single region, which may limit the generalizability of the findings to other parts of the country. Second, the HIV status of community PWUD, and the timelines of previous TB episodes were not captured in detail, representing a missed opportunity to further enrich our understanding of TB ACF among PWUD. To address these limitations,

we recommend that future studies consider incorporating comprehensive data on TB risk factors, better captured through mixed method research.

## Conclusion

Our study findings highlight peer provision of TB ACF services as a major facilitator in peer-led TB ACF, whereas barriers included a highly mobile PWUD population, inadequate adherence to IPC measures attributed to stigma and misconceptions and fear of withdrawal symptoms linked to use of anti-TB. Facilitators identified for targeted improvement in order to enhance peer led TB ACF were on the TB screening tool, integrated TB and MAT services and monetary incentive to peer PWUD.

**We recommend:**

1. Strengthen mobile diagnostic services by increasing their frequency and ensuring consistency, thus enabling broader coverage and early case detection and treatment among this highly mobile group.

2. Enhance the TB symptom screening tool by including a symptom independent test, such as a chest X-ray, as the symptoms of illicit drug use can mask and mimic TB symptoms making diagnosis challenging.

3. Address stigma and misconceptions through peer-led education and awareness campaigns that utilize audio-visual materials tailored to PWUD. This will promote adherence to IPC measures and create a more supportive environment for TB ACF activities.

4. Use anti-TB with minimal interactions with opiates or shorter TB treatment regimens to prevent withdrawal symptoms and improve adherence.

## Supporting information

**S1 File. Interview guides - English and Swahili.**
(PDF)

**S2 File. COREQ checklist.**
(PDF)

**S3 Fig. Factors influencing provision of TB ACF among PWUD.**
(TIF)

## Acknowledgments

We would like to thank all the study participants. We appreciate all the staff working at the Muhimbili, Mwananyamala and Temeke Medication Assisted Treatment clinics as well as the staff working at Mapambano ya Kifua Kikuu na Ukimwi Temeke (MUKIKUTE), STEPS Tanzania and Life and Hope Rehabilitation Organization (LHRO). We also thank the Swiss Tropical and Public Health Institute, National Institute for Medical Research and the Ifakara Health Institute.

## Author contributions

**Conceptualization:** Lilian Tina Minja, Liana Monica Minja, Doreen Pamba, Jerry Hella, Klaus Reither, Jennifer M. Belus.
**Data curation:** Lilian Tina Minja, Liana Monica Minja, Kilian Mlalama, Doreen Pamba.
**Formal analysis:** Lilian Tina Minja, Liana Monica Minja, Kilian Mlalama, Doreen Pamba, Klaus Reither, Jennifer M. Belus.
**Funding acquisition:** Lilian Tina Minja, Klaus Reither.

**Methodology:** Lilian Tina Minja, Kilian Mlalama, Doreen Pamba, Klaus Reither, Jennifer M. Belus.

**Project administration:** Lilian Tina Minja, Liana Monica Minja.

**Resources:** Jennifer M. Belus.

**Supervision:** Lilian Tina Minja, Jessie Mbwambo, Klaus Reither, Jennifer M. Belus.

**Validation:** Lilian Tina Minja, Liana Monica Minja, Kilian Mlalama, Doreen Pamba, Jerry Hella, Samwel Likindikoki, Cassian Nyandindi, Jessie Mbwambo, Klaus Reither, Jennifer M. Belus.

**Visualization:** Lilian Tina Minja, Liana Monica Minja, Kilian Mlalama, Doreen Pamba, Jerry Hella, Samwel Likindikoki, Cassian Nyandindi, Jessie Mbwambo, Klaus Reither, Jennifer M. Belus.

**Writing – original draft:** Lilian Tina Minja, Jennifer M. Belus.

**Writing – review & editing:** Lilian Tina Minja, Liana Monica Minja, Kilian Mlalama, Doreen Pamba, Jerry Hella, Samwel Likindikoki, Cassian Nyandindi, Jessie Mbwambo, Klaus Reither, Jennifer M. Belus.

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
