## [Decision Letter · Decision Letter 0]

19 Jan 2025

PONE-D-24-35095Facilitators, barriers, and strategies for the implementation of peer-led tuberculosis active case finding among people who use drugs in Dar es Salaam, Tanzania.PLOS ONE

Dear Dr. Minja,

Thank you for submitting your manuscript to PLOS ONE. After careful consideration, we feel that it has merit but does not fully meet PLOS ONE’s publication criteria as it currently stands. Therefore, we invite you to submit a revised version of the manuscript that addresses the points raised during the review process.

We look forward to receiving your revised manuscript.

Kind regards,

Harapan Harapan, MD, PhD

Academic Editor

PLOS ONE

Journal Requirements:

4. Please remove all personal information, ensure that the data shared are in accordance with participant consent, and re-upload a fully anonymized data set. 

Reviewers' comments:

Reviewer's Responses to Questions

**Comments to the Author**

1. Is the manuscript technically sound, and do the data support the conclusions?

Reviewer #1: Partly

Reviewer #2: Partly

Reviewer #3: Partly

Reviewer #4: Yes

2. Has the statistical analysis been performed appropriately and rigorously? 

Reviewer #1: No

Reviewer #2: Yes

Reviewer #3: No

Reviewer #4: I Don't Know

3. Have the authors made all data underlying the findings in their manuscript fully available?

Reviewer #1: Yes

Reviewer #2: No

Reviewer #3: No

Reviewer #4: Yes

4. Is the manuscript presented in an intelligible fashion and written in standard English?

Reviewer #1: Yes

Reviewer #2: Yes

Reviewer #3: Yes

Reviewer #4: Yes

5. Review Comments to the Author

Reviewer #1: The study entitled “Facilitators, barriers, and strategies for the implementation of peer-led tuberculosis active case finding among people who use drugs in Dar es Salaam, Tanzania” is a study that investigates an area that has not been extensively researched in Tanzania. However, some issues need clarification and improvements as detailed in the following sections:

Title: Well written

Abstract: well written

Introduction: Line 62-63: Citation is missing in this sentence.

Methods:

• Lines 102-104 (page 5), what are these three districts from which the participants were recruited? It is better to specify for clarity.

• Lines 108-112 (page 5) is not in its appropriate section. It should be included in a separate section for ethics approval. This section (of ethical approval) describes exactly how the informed consent was performed and obtained.

• Line 113 of the Checklist is not well filled out, as the authors claim. When counterchecked with the manuscript, the analysis section misses a lot of information.

• Lines 136-138: This sentence is odd as the qualitative interview is flexible and the tool may be modified during the interview to fit the merging themes. I suggest to be revised

• Line 146: Refusing to be recorded for a participant does not justify exclusion. You could have interviewed without recording and get useful information for your study. This sentence needs to be deleted or provide a good reason for such an abnormal exclusion.

• Line 156: How and who translated the data?

• Lines 162-166: This section needs extensive descriptions to understand how were the data analyzed (how were codes formed? how reliable were the codes? how did you arrive at the themes? Is there any supportive evidence of rigor in the analysis? This information is important as it informs the true findings in the results section.

Results:

• Lines 178: I suggest the facilitators be described separately from barriers. E.g. individual and social facilitators. This will improve the clarity of the findings.

• Line 295: Is this a facilitator or barrier? This is not clear. While counterchecking with Fig. 1, I found that this subtheme is written as an inadequate TB Screening Tool. This is not support what is written in narratives and the figure. Please correct this

Discussion:

Should also be structured in terms of individual and social facilitators discussed separately from barriers and then discussed logically and sequentially as appears in the results

Conclusion

What is the conclusion? The conclusion is missing the author only provided a recommendation.

Reviewer #2: Interesting article focusing on an important population. Given the importance of HIV as the number one risk factor for TB in Tanzania (estimated 20,000 are coinfected) and the increased rates of HIV in PWUD the lack of mentioning of HIV throughout the article needs addressed - a 2017 IBBS survey estimated the overall prevalence of HIV infection in PWUD in Tanzania was 8.7% (95% CI 6.5–10.9). Several studies conducted in Tanzania between 2010 and 2018 have reported that the prevalence of HIV infection among PWID ranges between 11.0 and 51.1% (Likindikoki, S.L., Mmbaga, E.J., Leyna, G.H. et al. Prevalence and risk factors associated with HIV-1 infection among people who inject drugs in Dar es Salaam, Tanzania: a sign of successful intervention?. Harm Reduct J 17, 18 (2020). From reviewing the questionnaires, for the Peer PWUD the HUV status was requested but not for the PWUD, which was a missed opportunity given the importance of this as a risk factor.

For the introduction section:

- please use the latest numbers from WHO Global TB report 2024, for example WHO Global TB report 2024 reports the incidence is now 183 per 100,000 for Tanzania.

- Can more information be given on the situation regarding PWUD in Tanzania; any updated IBBS or other data on rates of coinfection with HIV/HCV, what is the estimated burden of PWUD in Tanzania? Any data on the number of people who inject drugs, what is the legal situation regarding drug use in Tanzania (this may impact discussions around stigmatization and accessing care)

- With regards to current community active case finding activities more generally, can sputum samples be collected in the community or do people have to attend a health care facility? This knowledge can help understand the recommendation regarding access to diagnostic services/strengthening peer ACF if they can collect samples in the community.

For the Recruitment and sample selection:

- Was any consideration given to HIV status in the sample selection given prevalence and risk factor for TB?

- More information regarding the Peer PWUD would be helpful, such as what other peer activities do they perform (if any), were all of them involved in TB ACF? Reading later in the article it would seem that is the case but would be good to have the specifically stated when outlining the groups interviewed? Was any duration of time involved in ACF discussed, given that that was one of the questions asked?

- Was the interview location in a private room? It mentions being in the treatment center but did not say if it was in a more public/open space or private space.

Results:

- Were there any perceived issues or reflections regarding the population being interviewed being mainly male and the interviewees mainly female? Were the interviewees given any choice regarding who interviewed them?

- The previous TB history is important and more information regarding how many of the interviewees had a past history as this may have an impact on their responses regarding TB ACF and starting/staying in treatment, where peer PWUD more likely to have had a previous episode of TB? Did this help their work in engaging with PWUD? This data appears to have been recorded in both groups.

- The lack of information on HIV status makes the statements on IPC being identified as a risk factor for multiple episodes of TB when the main risk factor for TB is unknown (HIV). Additional risk factors such as malnutrition are als0 not mentioned. These multiple episodes of TB, was any time frame collected on this? Was this seen in Peer PWUD who had been doing it for a long period of time? What was the experience range of the peer PWUD?

- The discussions around usefulness of the TB screening questions when a positive symptom screen results in a negative test can not be fully understood without knowledge of the HIV status of the population - PLHIV are more likely to be paucibacillary and therefore more likely to have negative bacteriological samples regardless of PWUD status.

For Line 355 regarding the drop in services - are they already offering any TB screening/diagnostic services? It isn't clear if this is something that needs to start or can be expanded.

Reviewer #3: A review of the manuscript entitled “Facilitators, barriers, and strategies for the implementation of peer-led tuberculosis active case finding among people who use drugs in Dar es Salaam, Tanzania.”

1. (page 3, lines 58-60): Please use the most updated data on the number of TB cases in the Tanzanian population.

2. (page 3, lines 55-56): This sentence “Globally, tuberculosis (TB) is again the leading cause of death from a single infectious agent” will be more significant if it’s also supported by another relevant study. For example, “Risk factors for viral hepatitis in pulmonary tuberculosis patients undergoing treatment: A systematic review and meta-analysis”.

3. (page 5, lines 93-95): The motivation of the research is not clearly presented. What are the novelty aspects of this work?

4. (page 5, study setting and design): Why did the authors only recruit participants from three MAT clinics? In my view, a larger sample size will make the study findings stronger and more powerful. The current number of sample size seems somehow small to make the conclusion drawn from this study significant.

5. (page 8, lines 162-166): The data analysis part seems somewhat questionable as the authors did not give detailed information on how the data was analysed.

6. (page 18, lines 373-375): This sentence “Nonetheless, studies from COVID-19 suggest that uptake of wearing face masks can be enhanced by strengthening the belief of solidarity and promotion of positive social messages” also shares the same idea with a similar study by Hamdan et al entitled “Coping strategies used by healthcare professionals during COVID-19 pandemic in Dubai: A descriptive cross-sectional study”. Kindly include this reference to make the sentence stronger.

7. (page 19, lines 392-399): The last paragraph of the discussion part should be rewritten, especially in terms of highlighting the limitations of your study. The current wording is incorrect because the authors mentioned that this study has numerous strengths, and then in the elaboration they included the study’s weaknesses, which does not flow well with the previous statement of study strengths. Please revise.

Reviewer #4: The manuscript is well-written, and depicts the need to enhance TB services for PWUD to provide indiscriminate care and support. The discussion part is well-versed and highlights the ways forward. Although mobile TB van services are available, it is sad to know it only runs a couple times in a year, indicating the need to improve the integrated services. Overall, an informative and much needed manuscript for this specific population.

6. PLOS authors have the option to publish the peer review history of their article (what does this mean? ). If published, this will include your full peer review and any attached files.

**Do you want your identity to be public for this peer review?** For information about this choice, including consent withdrawal, please see our Privacy Policy .

Reviewer #1: No

Reviewer #2: No

Reviewer #3: No

Reviewer #4: No

---

## [Author Response · Author response to Decision Letter 0]

10 Mar 2025

Journal Requirements:

Response:

Thank you, we will ensure our manuscript meets PLOS ONE’s style requirements.

Response:

Thank you for your comment. We have included the full name of the IRB and ethics committee that approved the study as well as information on obtaining informed consent. We include the below information on Page 10 and 11, under the ethical considerations section in the revised manuscript:

The study received approvals from the National Health Research Ethics Review Committee and Ifakara Health Institute Institutional Review Board in Tanzania. The investigator welcomed the participant or legally acceptable representative (for illiterate participants) to participate. The investigator then provided the participant with an informed consent form, that provided information on the study, including its potential benefits and risks. Adequate time was provided to the participant to read and ask questions about the study and an assessment of understanding was conducted by the investigator to confirm the participant had understood the information provided. The language used was as non-technical as possible. Written informed consent was obtained from all study participants. In the case of illiteracy, informed oral consent was attested by an impartial witness and documented with the participant’s fingerprint.

Response:

Thank you for the comment regarding data sharing.

We recognize the importance of making data available for further research and enhance transparency. However, as some of the interview transcripts contain identifiable and sensitive information, making the data publicly available would compromise this confidentiality, and the assurance that we gave the participants that their data would remain confidential.

We now include the following data sharing description in the manuscript on Page27,

Data availability statement: Some interview transcripts include identifiable and sensitive information, and publicly releasing the data could compromise participant confidentiality. To protect participant privacy, the data generated and analyzed during this study is not publicly accessible. The data generated and analyzed can however be made available upon reasonable request, provided that the data sharing agreement is approved by the Medical Research Coordinating Committee in Tanzania. For data availability requests, please contact info@nimr.or.tz.

Response: Thank you. We have provided information on the restrictions on data sharing on Page 27 of the revised manuscript and provide contact address for data requests.

4. Please remove all personal information, ensure that the data shared are in accordance with participant consent, and re-upload a fully anonymized data set.

Response: Thank you. We have provided information on the restrictions on data sharing on Page 27 of the revised manuscript and provide contact address for data requests.

Reviewers' comments:

Reviewer's Responses to Questions

Comments to the Author

1. Is the manuscript technically sound, and do the data support the conclusions?

Reviewer #1: Partly

Reviewer #2: Partly

Reviewer #3: Partly

Reviewer #4: Yes

Response: Thank you for your valuable and comprehensive review. Following your valuable suggestions and comments, we provide detailed description on how our study was conducted in the revised manuscript, ensuring the rigorous conduct of our study is conveyed clearly in the manuscript. We provide our revised manuscript for your review and consideration for publication.

2. Has the statistical analysis been performed appropriately and rigorously?

Reviewer #1: No

Reviewer #2: Yes

Reviewer #3: No

Reviewer #4: I Don't Know

Response: Thank you for your comprehensive review. We have addressed comments and suggestions provided on the study analysis. Page 9 and 10 of our revised manuscript provides detailed information on how our study analysis was conducted reflecting the appropriate and rigorous aspects that were done. The data analysis section in the revised manuscript reads:

Data analysis: Thematic analysis was used to analyze the qualitative data using NVivo 12 software. The audio recordings were transcribed verbatim in Swahili before translation into English. The data was translated by a professional translator fluent in both the local language Swahili and English. The translator was familiar with the context of the study to ensure accuracy and cultural relevance. The translated transcripts and field notes were further double checked for accuracy by the research team (LTM, DP, KM) thus also allowing for familiarization of data. In order to develop initial draft codebooks to guide subsequent coding, LTM, DP and KM randomly selected one transcript from each of the three participant groups thus, each with a transcript to work on. Deductive codes were applied based on research questions and few others were emergent from the transcripts. Thereafter, the team met for a group review of the coded transcripts. During the review, the team extensively reviewed the codebooks. When discrepancies arose, agreements were reached through discussions. Thus, certain codes were deleted, merged and rephrased. The final codebooks were subsequently applied to the rest of the transcripts which were randomly divided among the team. Thereafter, codes were grouped into categories and themes were developed from categories with similar meaning.

3. Have the authors made all data underlying the findings in their manuscript fully available?

Reviewer #1: Yes

Reviewer #2: No

Reviewer #3: No

Reviewer #4: Yes

Response:

Thank you for the detailed review on data sharing.

We recognize the importance of making data available for further research and enhance transparency. However, as some of the interview transcripts contain identifiable and sensitive information, making the data publicly available would compromise this confidentiality and the assurance that we gave the participants that their data would remain confidential.

We now include the following data sharing description in the manuscript on Page 27

Data availability statement: Some interview transcripts include identifiable and sensitive information, and publicly releasing the data could compromise participant confidentiality. To protect participant privacy, the data generated and analyzed during this study is not publicly accessible. The data generated and analyzed can however be made available upon reasonable request, provided that the data sharing agreement is approved by the Medical Research coordinating Committee in Tanzania. For data availability requests, please contact info@nimr.or.tz________________________________________

4. Is the manuscript presented in an intelligible fashion and written in standard English?

Reviewer #1: Yes

Reviewer #2: Yes

Reviewer #3: Yes

Reviewer #4: Yes

Response:

Thank you for the comment, we appreciate your feedback.

5. Review Comments to the Author

Reviewer #1: The study entitled “Facilitators, barriers, and strategies for the implementation of peer-led tuberculosis active case finding among people who use drugs in Dar es Salaam, Tanzania” is a study that investigates an area that has not been extensively researched in Tanzania. However, some issues need clarification and improvements as detailed in the following sections:

Title: Well written

Abstract: well written

Response: Thank you for the positive feedback on the title and abstract.

Introduction: Line 62-63: Citation is missing in this sentence.

Response: Thank you for noting this and we apologize for missing to add the citation.

In our revised manuscript, we have added the below citation (Citation number 11 of the revised manuscript) to the sentence to ensure proper referencing and compliance with the guidelines.

WHO consolidated guidelines on tuberculosis systematic screening for tuberculosis disease; Module 2: Screening. Geneva, Switzerland: World Health Organization; 2021.

Methods:

• Lines 102-104 (page 5), what are these three districts from which the participants were recruited? It is better to specify for clarity.

Response:

Thank you for your suggestion.

We have specified the three districts from which the participants were recruited on Page 6 of the revised manuscript as shown below:

Participants were recruited from three different MAT clinics located in Ilala, Kinondoni and Temeke districts.

• Lines 108-112 (page 5) is not in its appropriate section. It should be included in a separate section for ethics approval. This section (of ethical approval) describes exactly how the informed consent was performed and obtained.

Response:

We agree that the information regarding informed consent would be more appropriately placed under the ethics approval section. We have moved the content to the relevant section under Ethical considerations on page 10 and 11 of the revised manuscript.

• Line 113 of the Checklist is not well filled out, as the authors claim. When counterchecked with the manuscript, the analysis section misses a lot of information.

Response: We apologize for the oversight in the COREQ checklist. We have revised the analysis section in the manuscript and provide detailed information to ensure that all relevant information is included and accurately reflects the content. An updated COREQ checklist has been appended.

The data analysis section (Page 9 and 10) in the revised manuscript reads:

Data analysis: Thematic analysis was used to analyze the qualitative data using NVivo 12 software. The audio recordings were transcribed verbatim in Swahili before translation into English. The data was translated by a professional translator fluent in both the local language Swahili and English. The translator was familiar with the context of the study to ensure accuracy and cultural relevance. The translated transcripts and field notes were further double checked for accuracy by the research team (LTM, DP, KM) thus also allowing for familiarization of data. In order to develop initial draft codebooks to guide subsequent coding, LTM, DP and KM randomly selected one transcript from each of the three participant groups thus, each with a transcript to work on. Deductive codes were applied based on research questions and few others were emergent from the transcripts. Thereafter, the team met for a group review of the coded transcripts. During the review, the team extensively reviewed the codebooks. When discrepancies arose, agreements were reached through discussions. Thus, certain codes were deleted, merged and rephrased. The final codebooks were subsequently applied to the rest of the transcripts which were randomly divided among the team. Thereafter, codes were grouped into categories and themes were developed from categories with similar meaning.

• Lines 136-138: This sentence is odd as the qualitative interview is flexible and the tool may be modified during the interview to fit the merging themes. I suggest to be revised

Response: As you correctly pointed out, qualitative interview guides are inherently flexible, and this was carefully considered during the study. We have revised the section on Page 8 and as shown below to clarify how the interviews were conducted, emphasizing the flexibility of the interview process to accommodate emerging themes.

The interview guides were piloted on two peer PWUD and one MAT client. Feedback from pilot interviews indicated that the questions were well understood and no substantive changes were needed in the interview guides. However, during subsequent interviews, questions in the interview guides were rephrased

---

## [Decision Letter · Decision Letter 1]

16 Apr 2025

Facilitators, barriers, and strategies for the implementation of peer-led tuberculosis active case finding among people who use drugs in Dar es Salaam, Tanzania.

PONE-D-24-35095R1

Dear Dr. Minja,

We’re pleased to inform you that your manuscript has been judged scientifically suitable for publication and will be formally accepted for publication once it meets all outstanding technical requirements.

Kind regards,

Harapan Harapan, MD, PhD

Academic Editor

PLOS ONE

Additional Editor Comments (optional):

Reviewers' comments:

Reviewer's Responses to Questions

**Comments to the Author**

1. If the authors have adequately addressed your comments raised in a previous round of review and you feel that this manuscript is now acceptable for publication, you may indicate that here to bypass the “Comments to the Author” section, enter your conflict of interest statement in the “Confidential to Editor” section, and submit your "Accept" recommendation.

Reviewer #3: All comments have been addressed

Reviewer #4: All comments have been addressed

2. Is the manuscript technically sound, and do the data support the conclusions?

Reviewer #3: Yes

Reviewer #4: Yes

3. Has the statistical analysis been performed appropriately and rigorously? 

Reviewer #3: Yes

Reviewer #4: I Don't Know

4. Have the authors made all data underlying the findings in their manuscript fully available?

Reviewer #3: Yes

Reviewer #4: Yes

5. Is the manuscript presented in an intelligible fashion and written in standard English?

Reviewer #3: Yes

Reviewer #4: Yes

6. Review Comments to the Author

Reviewer #3: I commend the authors for addressing my concerns. Now this manuscript has improved a lot and is ready for further consideration.

Reviewer #4: Thank you for revising the manuscript, and responding to the comments. Informative and much needed voice of the specific population.

7. PLOS authors have the option to publish the peer review history of their article (what does this mean? ). If published, this will include your full peer review and any attached files.

**Do you want your identity to be public for this peer review?** For information about this choice, including consent withdrawal, please see our Privacy Policy .

Reviewer #3: No

Reviewer #4: No

---

## [Editor Report · Acceptance letter]

PONE-D-24-35095R1

PLOS ONE

Dear Dr. Minja,

I'm pleased to inform you that your manuscript has been deemed suitable for publication in PLOS ONE. Congratulations! Your manuscript is now being handed over to our production team.

Kind regards,

on behalf of

Dr. Harapan Harapan

Academic Editor

PLOS ONE